# Development of a Highly Specific Monoclonal Antibody-Based Sandwich ELISA for Rapid Detection of Porcine Circovirus Type 3

**DOI:** 10.3390/v17101340

**Published:** 2025-10-05

**Authors:** Zhen Li, Jiaying Zhao, Ang Tian, Hao Wu, Huanchun Chen, Yunfeng Song

**Affiliations:** 1State Key Laboratory of Agricultural Microbiology, Huazhong Agricultural University, Wuhan 430070, China; 2College of Veterinary Medicine, Huazhong Agricultural University, Wuhan 430070, China; 3Key Laboratory of Development of Veterinary Diagnostic Products, Ministry of Agriculture and Rural Affairs of China, Wuhan 430070, China

**Keywords:** porcine circovirus 3 (PCV3), capsid protein, monoclonal antibody (mAb), double antibody sandwich ELISA (DAS-ELISA)

## Abstract

Porcine circovirus type 3 (PCV3), initially identified in the United States in 2016, is associated with multisystemic inflammation, myocarditis, reproductive failure in sows, and growth retardation in piglets, posing a significant economic threat to the swine industry. In this study, prokaryotic-expressed recombinant PCV3 Cap protein was used to immunize mice and rabbits. A monoclonal antibody (mAb 4G1) was generated through hybridoma technology, targeting a novel linear epitope (^37^DYYDKK^42^) within the first β-sheet of the Cap structure. This epitope exhibits high conservation (99.35%, 1239/1247) based on sequence alignment analysis, and residues 39 and 42 are critical residues affecting mAb binding. Subsequently, using rabbit polyclonal antibody (pAb) as the capture antibody and mAb 4G1 as the detection antibody, a double antibody sandwich ELISA (DAS-ELISA) method was developed. The assay demonstrates a cut-off value of 0.271, a detection limit for positive pig serum is 1:800, and shows no cross-reactivity with other swine pathogens. Intra- and inter-assay coefficients of variation were <10%, with a linear detection range for Cap protein down to 3.4 ng/mL. The coincidence rate between the DAS-ELISA and qPCR was 93.33% (70/75) for PCV3 detection in serum, with a kappa value of 0.837. This study establishes a simple, sensitive, and operationally efficient DAS-ELISA and provides a reference for monitoring PCV3 infection in swine herds.

## 1. Introduction

Porcine circovirus type 3 (PCV3) is a single-stranded, closed circular DNA virus encapsulated by a capsid, which belongs to the genus Circovirus within the family Circoviridae [1]. PCV3 was initially detected through metagenomic sequencing in the United States in 2016 and has since achieved global distribution [2,3]. PCV3 has been a concern since its discovery, given the clinical and economic consequences caused by PCV2 [4,5]. Accumulating evidence from clinical case reports and experimental infections has established PCV3’s association with multisystemic inflammatory syndrome, myocarditis, reproductive disorders in sows, and growth retardation in piglets, which demonstrate significant economic impacts on the swine industry [6,7,8,9]. In addition, PCV3 can be transmitted both horizontally and vertically. The infection leads to persistent viremia, with intermittent viral shedding observed in feces, nasal secretions, semen, and colostrum [10,11]. Moreover, PCV3 nucleic acid can be widely detected in various tissues, including the heart, liver, spleen, lungs, kidneys, intestines, and lymph nodes [12]. Beyond domestic pigs and wild boars, PCV3 exhibits a broad host range encompassing cattle, donkeys, dogs, foxes, red deer, roe deer, laboratory mice, ticks, and mosquitoes [1,13]. Molecular epidemiology indicates a high prevalence of PCV3 across Chinese pig farms, particularly in sows, with frequent co-infections involving pathogens such as PCV2, porcine reproductive and respiratory syndrome virus (PRRSV), porcine parvovirus (PPV), Torque teno sus virus (TTSuV), and Mycoplasma suis [14,15,16,17,18,19]. Notably, PCV3 may possess immunosuppressive properties that enhance susceptibility to secondary infections [20,21].

PCVs share a similar viral structure characterized by an icosahedral capsid that encapsulates a single-stranded circular DNA genome [22]. The genome contains two major open reading frames, encoding the capsid protein (Cap) and replication-associated protein (Rep), respectively [23]. As the sole immunogenic structural protein, PCV3 Cap mediates viral entry and immune recognition [24]. The PCV3 Cap consists of 214 amino acids. Its amino acid (aa) sequence shows high conservation, but shares only 24–37% homology with PCV1, PCV2, and PCV4 [25]. Phylogenetic and selection pressure analyses revealed low plasticity and less efficient host-induced natural selection of PCV3 [26]. Although several major mutation sites of the Cap protein have been identified, the absence of a stable cell line hinders pathogenesis research [27]. Therefore, establishing a detection method is imperative for implementing herd-level surveillance of PCV3 infection.

It is reported that current PCV2 vaccines fail to confer cross-protection against PCV3 [28]. However, PCV3 detection has been facilitated by the development of multiple diagnostic methodologies, including PCR, quantitative fluorescence PCR (qPCR), recombinase polymerase amplification (RPA), loop-mediated isothermal amplification (LAMP), enzyme-linked immunosorbent assay (ELISA), in situ hybridization (ISH), and immunohistochemistry (IHC) [11,29]. Among these, qPCR and ELISA have emerged as predominant techniques in swine disease diagnostics. While qPCR enables multi-targeted pathogen nucleic acid detection, ELISA serves as a robust serological approach for antigen and antibody [30]. To date, indirect ELISA and blocking ELISA for PCV3 detection have been documented [31,32]. However, an antigen-specific double antibody sandwich ELISA (DAS-ELISA) remains undeveloped. The DAS-ELISA, typically based on monoclonal antibody (mAb), represents a highly sensitive and specific method for antigen detection [33]. A developed PCV2-specific DAS-ELISA has been successfully employed to quantify the effective antigen content in vaccines [34]. Consequently, the development of a sandwich ELISA is of great significance for the clinical detection of PCV3.

In the present study, PCV3 Cap protein lacking nuclear localization signal (NLS) was expressed in *Escherichia coli* (*E. coli*) via a prokaryotic expression system. The recombinant protein was utilized to immunize mice, generating a mAb against PCV3 Cap through hybridoma technology and characterized. This mAb was then paired with rabbit polyclonal antibodies (pAb) to establish a DAS-ELISA, providing a critical tool for advancing PCV3 diagnostics.

## 2. Materials and Methods

### 2.1. Cells, Vectors, and Animals

HEK293T cells (ATCC, Manassas, VA, USA) were cultured in Dulbecco’s Modified Eagle’s Medium (DMEM, Servicebio, Wuhan, China) supplemented with 10% fetal bovine serum (FBS, Yeasen, Shanghai, China) and penicillin/streptomycin (Biosharp, Hefei, China) at 37 °C in a humidified incubator supplied with 5% CO_2_. SP2/0 myeloma cells were cultured in RPMI 1640 medium (Servicebio, China) supplemented with 20% FBS and penicillin/streptomycin at 37 °C in a humidified incubator supplied with 5% CO_2_. The pET-30a(+) vector (BioVector NTCC, Beijing, China) was used for prokaryotic expression of the PCV3 Cap. The pET-28a-SUMO vector (BioVector NTCC, China) was used for expression of the truncated Cap protein. The pCAGGS-HA vector (BioVector NTCC, China) was used for the verification and characterization of mAb. BALB/*c* mice (6-week-old, SPF grade) and New Zealand White rabbits (2.5 kg) were purchased from the Experimental Animal Center of Huazhong Agricultural University (Wuhan, China).

### 2.2. Expression and Purification of PCV3 Cap Protein

The codon-optimized PCV3 *cap* gene sequence was synthesized by Tsingke Biotechnology (Wuhan, China). A truncated *cap* fragment lacking NLS (ΔN33Cap) was amplified using primer pair (forward primer: 5′-GGAATTCCATATGACCGCGGGTACCTACTACACC-3′; reverse primer: 5′-CCCAAGCTTCAGCACGCTTTTGTAACGAATCC-3′). The *cap* (truncated NLS) and pET-30a were digested with restriction enzymes *Nde* I and *Hind* III (TaKaRa Bio, Dalian, China), then ligated using T4 DNA ligase (TaKaRa Bio, China) to construct the recombinant plasmid pET-30a-ΔN33Cap. After the sequence was verified correctly, the plasmid was transformed into *E. coli* BL21(DE3) competent cells (Tsingke, Beijing, China). Protein expression was induced with 2 mM IPTG at 37 °C with 180 r/min shaking for 4 h. Bacterial cells were lysed, and inclusion bodies were collected at 12,000× *g* for 30 min at 4 °C. The inclusion bodies were solubilized in lysis buffer (25 mM Tris-HCl, 300 mM NaCl, 8 M urea, 1 mM DTT, pH 8.0). Protein purification was performed using Ni-NTA affinity chromatography (Bio-rad, Hercules, CA, USA). The eluted recombinant ΔN33Cap protein refolding was achieved by gradient dialysis against decreasing urea concentrations, then analyzed by 12% SDS-polyacrylamide gel electrophoresis (SDS-PAGE) and further verified by Western blot using anti-His tag monoclonal antibody (Proteintech, Wuhan, China).

### 2.3. Production of Anti-PCV3 Cap Antibodies

Purified ΔN33Cap protein (1 mg/mL) was emulsified in a 1:1 ratio with Freund’s complete adjuvant (Biodragon, Suzhou, China). BALB/*c* mice were immunized by subcutaneous multi-point injection (200 μL) in the cervical dorsal region. Two weeks later, two booster immunizations were given at 2-week intervals using Freund’s incomplete adjuvant (Biodragon, China) via the same injection protocol. Three weeks after the third immunization, mice received an intraperitoneal antigen boost. Three days post-boost, splenocytes were isolated and fused with SP2/0 myeloma cells using polyethylene glycol (PEG-1500)-mediated fusion (Sigma-Aldrich, St. Louis, MO, USA). Hybridoma cells were cultured in HAT medium and followed by HT medium (Sigma-Aldrich, St. Louis, MO, USA) for initial screening. Cell culture supernatants were screened by indirect ELISA to identify positive hybridoma clones. Positive clones underwent three rounds of subcloning via limiting dilution, with repeated indirect ELISA validation to select stable antibody-secreting clones. For ascites production, mice were pre-stimulated with a special adjuvant for ascites (Biodragon, China) for two weeks prior to intraperitoneal injection of 1 × 10^6^ hybridoma cells. Ascitic fluid was collected after approximately 10 days, clarified by centrifugation (1000× *g*, 10 min, 4 °C), and further purified using Protein A/G affinity chromatography (Beyotime, Shanghai, China) according to the manufacturer’s protocol.

Rabbits were immunized using the mouse immunization protocol with doubled antigen dosage. Serum was isolated from blood collected via cardiac puncture. Rabbit pAb were subsequently purified from the rabbit serum using the same purification method as described above.

### 2.4. Characterization of Anti-PCV3 Cap mAb and pAb

The purified mAb and pAb were first verified by SDS-PAGE. Then mAb and pAb were separated with SDS-PAGE and transferred to polyvinylidene difluoride (PVDF) membranes (Millipore, Burlington, MA, USA) for Western blot. The membranes were incubated directly with HRP-conjugated goat anti-mouse (for mAb) or HRP-conjugated goat anti-rabbit (for pAb) (Bio-rad, USA) to validate antibody specificity. Antibody–antigen binding was confirmed by parallel Western blot using mAb/pAb as primary antibodies, followed by corresponding HRP-conjugated secondary antibodies.

For indirect immunofluorescence (IFA) analysis, HEK-293T cells were cultured in 24-well plates until reaching 70% confluence. Transfection with 0.5 μg of pCAGGS-HA-Cap was performed using a lipofectamine reagent (Lipo3000, Yeasen, China). After 12 h, the cells were fixed with pre-cooled methanol for 15 min, followed by blocking with 1% bovine serum albumin (BSA) for 1 h. Then, they were incubated with mAb and pAb, respectively, and subsequently with corresponding fluorescently labeled secondary antibodies (DyLight488, Abbkine, Wuhan, China). Finally, cells were incubated with DAPI for 10 min to stain the cell nuclei. Fluorescence images were captured using a fluorescence microscope.

### 2.5. Fine Mapping of Epitope Targeted by mAb

To characterize the minimal linear epitope recognized by mAb, the full-length PCV3 Cap was divided into three fragments (residues 1–70, 71–124, and 125–214) named CapN, CapM, and CapC. Each fragment was cloned into the vector pET-30a and expressed in *E. coli* BL21(DE3) for recombinant truncated protein production. Truncated proteins were analyzed via Western blot to identify those reactive with the mAb. For positive fragments, truncation analysis was performed by sequential deletions at both termini, followed by re-expression in pET-28a-SUMO. The minimal reactive fragment that lost binding affinity upon removal of any terminal residue was defined as the core epitope. In parallel, all available PCV3 genome sequences uploaded to GenBank as of 2024 (Appendix A) were retrieved. Multiple sequence alignment was performed using MEGA11 to assess the conservation of the core epitope. The predicted Cap protein structure was then visualized with PyMOL, and the location of the core epitope was mapped and labeled within the structural model. Key residues within this epitope were systematically mutated to alanine (A). The impact of these mutations on mAb binding was validated using IFA and Western blot. The primer information is displayed in Appendix A.

### 2.6. Optimization of DAS-ELISA Procedure

The DAS-ELISA procedure was systematically optimized using purified pAb as the capture antibody and HRP-conjugated mAb (HRP-mAb) as the detection antibody. The mAb 4G1 was conjugated with HRP using the periodate method, according to the manufacturer’s instructions (Sangon, Himi, Japan). Beginning with checkerboard titration using two-fold serial dilutions to determine optimal pAb coating concentrations (8–0.25 μg/mL) and HRP-mAb working dilutions (1:100–1:12,800). The optimization process included sequential testing of: coating conditions (coating at 37 °C for 0.5–2 h and coating at 4 °C for 12 h); blocking parameters (1%, 2.5%, 5% BSA/skim milk, with incubation durations tested from 0.5 to 2 h at 0.5 h intervals); serum incubation durations (incubated from 15 min to 2 h at 15 min intervals) and HRP-mAb incubation durations (incubated from 15 min to 2 h at 15 min intervals); and TMB substrate reaction durations (2, 4, 6, 8, and 10 min). Each optimal step was ultimately determined based on maximizing the positive-to-negative serum OD_450_ ratio (P/N value). The positive and negative sera were gifted from Wuhan Keqian Biology Co., Ltd. (Wuhan, China).

### 2.7. Performance Verification of DAS-ELISA

The developed DAS-ELISA was performed to test 25 PCV3 antigen-negative serum samples. The cut-off value was the mean OD_450_ value plus three standard deviations (SD). Specificity was evaluated against cross-reactive pathogens that were collected and stored in our lab, including: *Escherichia coli* (EC), *Streptococcus suis* (SS), *Haemophilus parasuis* (HPS), *Salmonella* (SM), porcine dltacoronavirus (PDCoV), porcine epidemic diarrhea virus (PEDV), Zika virus (ZIKV), pseudorabies virus (PRV), Japanese encephalitis virus (JEV), vesicular stomatitis virus (VSV), and porcine rotavirus (PoRV), with parallel PCV3-positive/negative controls. Sensitivity was determined by testing serially two-fold diluted PCV3-positive sera (initial dilution 1:100). Intra-assay variation was assessed using four serum samples analyzed in quadruplicate on one plate; inter-assay variation was evaluated across four independent plates. Coefficients of variation (CVs) were calculated from the mean ($\bar{X}$) and standard deviation (SD) to verify assay precision. For quantitative analysis, a two-fold serial dilution series of purified ΔN33Cap protein (439.5–3.4 ng/mL) was analyzed. The standard curve was plotted with ΔN33Cap concentration as the abscissa and OD_450_ as the ordinate, demonstrating linear detection capability. To evaluate the performance of the DAS-ELISA, a comparative analysis was performed on 75 routine serum samples collected from healthy pigs on a farm in Guangzhou. All samples were first tested for PCV3 nucleic acid by qPCR and subsequently for PCV3 antigen by DAS-ELISA. The agreement between the two methods, quantified by the kappa value, was calculated using Microsoft Excel.

### 2.8. Statistical Analysis

Data statistical analysis was conducted using Microsoft Excel 2019. The distribution of negative and positive sample data was visualized through bar charts, with the changing trend of P/N values visualized through line charts. The mean and variance were used to characterize the central tendency and quantify dispersion.

## 3. Results

### 3.1. PCV3 Cap Antigen Preparation

The ΔN33Cap protein was successfully expressed in *E. coli* BL21(DE3) using a prokaryotic expression system. SDS-PAGE analysis showed that the purified protein revealed a distinct band at approximately 22 kDa, consistent with the predicted molecular weight (Figure 1A). Subsequent Western blot analysis using an anti-6×His tag monoclonal antibody also confirmed the presence of the ΔN33Cap protein at the expected size (Figure 1B).

### 3.2. Preparation and Characterization of mAb and pAb Against PCV3 Cap

Mice were immunized three times with purified ΔN33Cap protein, eliciting antibody titers of 1:200,000 (Figure 2A and Appendix A), qualifying spleen cells for fusion. Rabbits immunized with the same antigen elicited titers of 1:100,000, which could be used for pAb purification (Figure 2A and Appendix A). The mAb 4G1 (from ascites) and pAb (from serum) purified by Protein A/G chromatography showed high purity on SDS-PAGE, with distinct bands at approximately 55 kDa (heavy chain) and 25 kDa (light chain), consistent with the molecular weights of immunoglobulin components (Figure 2B,C). Western blot confirmed chain integrity and specific binding to ΔN33Cap (Figure 2B,C, Appendix A). Purified mAb and pAb titers reached 1:1,638,400 (Figure 2D). IFA revealed nuclear-localized HA-Cap in HEK-293T cells, with specific green fluorescence using mAb/pAb as primary antibodies (Figure 2E).

### 3.3. Fine Mapping of Epitope Targeted by mAb Against PCV3 Cap

To identify the minimal linear epitope recognized by the mAb, a series of truncated Cap proteins were generated (Figure 3A). Western blot revealed the minimal binding epitope as the hexapeptide ^37^DYYDKK^42^ (Figure 3B), located within the first β-sheet of the N-terminal domain of the Cap protein (Figure 4B). Statistical analysis of the amino acid sequence alignment of PCV3 Cap reveals that the hexapeptide segment is highly conserved. Systematic alanine-scanning mutagenesis further validated critical residues within this epitope. Western blot results demonstrated that substitutions at positions Y38A, Y39A, and K41A completely abolished mAb binding, as evidenced by the absence of specific bands (Figure 4C). In parallel, IFA revealed that mutations targeting the conformational epitope (Y39A and K42A) also disrupted mAb recognition, with no detectable green fluorescence observed in the nuclei (Figure 4D).

### 3.4. Optimal DAS-ELISA Procedure

A DAS-ELISA was developed using the prepared pAb as the capture antibody and HRP-mAb as the detection antibody. Each procedural step was systematically optimized, with the maximum positive-to-negative (P/N) ratio serving as the criterion for determining optimal conditions. The optimal reaction conditions were performed as follows: First, the pAb was diluted to 1 μg/mL in carbonate-bicarbonate buffer (pH 9.6), then used to coat the plates (100 μL/well) at 4 °C for 12 h (Figure 5A,B). Second, the plates were blocked with 5% BSA solution (200 μL/well) at 37 °C for 2 h (Figure 5C). Subsequently, serum was incubated in the plates (100 μL/well) at 37 °C for 1.5 h (Figure 5D), followed by the addition of HRP-mAb diluted 1:800 (100 μL/well) at 37 °C for 30 min (Figure 5A,E). Finally, TMB substrate (100 μL/well) was added for color development at room temperature for 6 min (Figure 5F). Between each step, the plates were washed three times with 200 μL PBST (5 min per wash). The reaction was terminated by adding 100 μL of 2 mol/L H_2_SO_4_, and the absorbance was measured utilizing an automatic microplate reader at 450 nm. The test data information is provided in Appendix A.

### 3.5. Evaluation of DAS-ELISA

The optimized DAS-ELISA was employed to evaluate 25 PCV3 antigen-negative serum samples. The mean OD_450_ value ($\bar{X}$) was determined as 0.178 with a standard deviation (SD) of 0.0311. The cut-off value was calculated as $\bar{X}$ + 3SD = 0.271 (Figure 6A). Method validation demonstrated exceptional specificity, as no cross-reactivity was observed with EC, SS, HPS, SM, PDCoV, PEDV, ZIKV, PRV, JEV, VSV, PoRV, and PCV2 (Figure 6B). The assay exhibited high sensitivity, consistently detecting positive signals in four distinct serum samples even at 1:800 dilutions (Figure 6C). A linear standard curve of DAS-ELISA was obtained with the detection range of 3.4–439.5 ng/mL, the linear equation was y = 0.004x + 0.0787 with R^2^ = 0.9979 (Figure 6D). Repeatability assessments revealed excellent precision, with both intra-assay and inter-assay coefficients of variation (CV) remaining below 10% across replicate experiments (Table 1 and Table 2). The test data information is provided in Appendix A. A total of 75 serum samples collected from pig farms were tested simultaneously by qPCR and DAS-ELISA. As shown in Table 3, both methods detected 51 positive samples and 19 negative samples. Discordant results were observed in five samples: three samples were DAS-ELISA-negative but qPCR-positive, while two samples were DAS-ELISA-positive but qPCR-negative. The overall agreement between the two methods was 93.33% (70/75), with a kappa coefficient (κ) of 0.837, indicating substantial agreement (Table 3). The test data information is provided in Appendix A.

## 4. Discussion

Since its initial report, PCV3 has attracted significant attention and has emerged as a pathogen of global concern [35]. The virus is pathogenic, with histological lesions characterized by prominent lymphohistiocytic arteritis and periarteritis, frequently associated with myocarditis and multisystemic inflammation [29]. PCV3 infection contributes to reproductive failure in sows and growth retardation in piglets, resulting in substantial economic losses for the swine industry [9]. The detection of PCV3 DNA in multiple host species underscores its broad transmission potential, which undoubtedly compounds disease control challenges [36]. Furthermore, no effective vaccine against PCV3 is currently available. Although successful isolation of PCV3 using primary cells has been reported, the inherent difficulties in procuring and maintaining these cell lines limit their suitability for sustained experimentation [37,38]. Consequently, the precise pathogenic mechanisms underlying PCV3 infection remain incompletely understood.

As the core structural protein of PCV3, the Cap protein is not only an essential component for viral assembly and host cell entry but also represents a critical molecule for its immunogenicity and as a diagnostic target [22]. The generation of mAbs against the PCV3 Cap protein could provide a highly specific tool for elucidating the virus’s pathogenic mechanisms. Furthermore, the high homogeneity and reproducibility inherent to mAbs establish a crucial technical foundation for developing highly sensitive and specific diagnostic assays [39]. Li et al. utilized the pET-28a vector to express the full-length Cap protein in *E. coli* and subsequently screened a mAb (1H1) capable of specifically recognizing PCV3 Cap, which was successfully applied in IHC [40]. In the present study, BALB/*c* mice were immunized with prokaryotically expressed and purified recombinant ΔN33Cap protein. Employing hybridoma technology, we successfully isolated four stable mAb-producing hybridoma cell lines. However, an additivity ELISA and subsequent sequencing analysis revealed that all four mAbs recognize the same epitope (Appendix A). This limited epitope diversity is likely attributable to insufficient protein renaturation, potentially resulting in an incomplete native conformation that presents too few immunogenic epitopes to stimulate a broader antibody repertoire.

Antigenic epitopes represent the critical regions within an antigen molecule that mediate specific binding to antibodies or T-cell receptors. Epitope identification facilitates the screening of species- or type-specific epitopes, thereby minimizing cross-reactivity with other pathogens and ensuring the specificity of diagnostic reagents, such as ELISA kits [41]. Several antigenic epitopes within the PCV3 Cap protein have been identified. Jiang et al. utilized the pGEX-6P-1 vector to express a Cap fragment (45–214 aa) in *E. coli*, enabling the screening of mAbs targeting the epitopes ^57^NKPWH^61^, ^140^KHSRYFT^146^, and ^161^QSLFFF^166^ [42]. Employing the pET-32a vector to express a codon-optimized full-length Cap protein, Wang et al. isolated mAb 7E3, which recognizes the epitope ^110^DLDGAW^115^ [32]. Chang et al. expressed a Cap fragment spanning residues 110–214 using pET-24a and identified mAb CCC160, which specifically binds to residues 110–160 [43]. Notably, this 110–160 aa region encompasses both the 110–115 aa and 140–146 aa epitopes reported previously. Yang et al. employed pET-28a to express an NLS-deleted Cap protein in *E. coli*, generating mAb 2D6 recognizes ^47^NVISVGT^53^ [44]. In the present study, we identified the mAb 4G1, which specifically recognizes the epitope ^37^TYYTKK^42^. Furthermore, based on the predicted 3D structure of the Cap protein, ElliPro (http://tools.iedb.org/ellipro/; accessed on 15 May 2024) predicted a high-scoring B-cell epitope ^34^TAGTYYTKKY^43^ that encompasses this sequence (Appendix A). Importantly, the epitope ^37^TYYTKK^42^ is a novel linear B-cell epitope identified herein, which provides new insights into the structural and functional studies of the PCV3 Cap protein.

Although PCV3 has been confirmed to infect pigs across all production stages, including both healthy and abnormal pigs [11], the kinetics underlying farm-level infection dynamics and disease manifestation remain poorly understood. This knowledge gap poses a significant challenge to effective PCV3 control. Consequently, establishing a rapid and sensitive detection method for PCV3 infection is imperative. While ISH and IHC enable high-resolution spatial localization of the virus, their clinical utility is hampered by low availability, high costs, and impracticality for population-level screening. Conversely, qPCR facilitates group testing but requires nucleic acid extraction and sophisticated instrumentation, resulting in prolonged turnaround times. In contrast, the antibody-sandwich ELISA format offers operational simplicity, minimal equipment dependency, and enhanced specificity through dual antibody recognition, along with high sensitivity for antigen detection [33]. Notably, no antigen-detection assays for PCV3 have been reported to date. This study addresses this gap by innovatively developing a sandwich ELISA using rabbit pAb as capture agents and mouse mAb as detectors. This design leverages the broad antigen-binding capacity of pAb while ensuring specificity through mAb, achieving a detection limit of 3.4 ng/mL for the PCV3 Cap protein.

Given that PCV3 is detectable in multiple tissues of infected pigs and sporadically in oral fluids, feces, nasal swabs, semen, and colostrum—coupled with evidence of persistent viremia post-infection [8,29]. Therefore, serum was selected as the detection target in this study to maximize detection efficiency. To evaluate the performance of the DAS-ELISA, we tested randomly collected serum samples from clinically healthy pigs. In the absence of commercial kits for direct comparison, samples were initially screened using qPCR to detect PCV3 nucleic acid, followed by antigen detection using the DAS-ELISA, to assess the concordance between the two methods. The comparison with qPCR, a commonly used diagnostic tool in swine farms, serves as a preliminary indication of the detection capability of the DAS-ELISA. This comparison was not to establish a direct correlation but to use qPCR, the most widely adopted and sensitive method for detecting PCV3 genetic material, as a benchmark for an initial clinical performance evaluation of our assay [14,45]. A general trend was observed wherein samples with lower Ct values (indicating higher viral DNA load) tended to yield higher OD_450_ values in the DAS-ELISA, suggesting a positive association between viral genome detection and Cap protein antigen levels. However, PCV3 can be transmitted both horizontally and vertically, while its infection, replication, nucleic acid degradation, and antigen degradation under field conditions exhibit a cyclical and recurrent pattern [46]. Moreover, the peaks of viremia (DNA detection) and antigenemia (protein detection) may not occur simultaneously during the course of infection [47]. Consequently, the relationship between nucleic acid and antigen levels is difficult to fully elucidate using cross-sectional clinical serum samples alone. The observations derived from this sample set may not be fully representative or generalizable, and the primary aim of this analysis was to validate the DAS-ELISA itself. We did observe that a formal and rigorous correlation analysis between Ct and OD values would require longitudinal monitoring in experimental PCV3 infection studies. For field-level investigations, large-scale testing of clinical samples is needed, and the relationship between these values may be influenced by factors such as the age and health status of the pigs. However, such an analysis first requires establishing the kinetics of PCV3 nucleic acid and antigen presence through experimental infection to define the detection window. This will be the main objective of our follow-up work. In summary, these findings provide a powerful tool for exploring the dynamics of antigen during PCV3 replication and offer a valuable reference for future diagnostic development.

## 5. Conclusions

In summary, our research immunized mice with prokaryotic expression of PCV3 Cap protein and screened out a mAb 4G1 that recognizes a novel linear epitope ^37^DYYDKK^42^ with high conservation, and identified that residues 39 and 42 are critical for mAb binding. By combining mAb 4G1 with rabbit pAb, a double antibody sandwich ELISA was established, which has good sensitivity and specificity, and provides a new method for the detection of PCV3 infection.

## Figures and Tables

**Figure 1 viruses-17-01340-f001:**
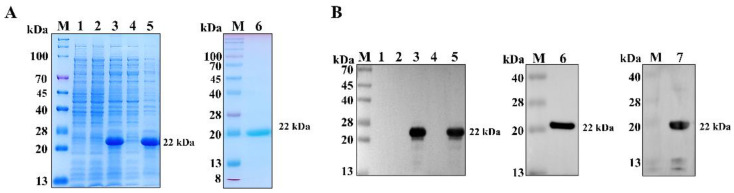
Expression and purification of PCV3 ΔN33Cap protein. SDS-PAGE (**A**) and Western blot (**B**) analysis of ΔN33Cap protein expression and purification. Lane M: Protein marker; lane 1: IPTG-induced *E. coli* BL21(DE3) lysate with empty vector pET-30a; lane 2: uninduced *E. coli* BL21(DE3) lysate with pET-30a-ΔN33Cap; lane 3: IPTG-induced *E. coli* BL21(DE3) lysate with pET-30a-ΔN33Cap; lane 4: supernatant of IPTG-induced *E. coli* BL21(DE3) lysate with pET-30a-ΔN33Cap; lane 5: inclusion bodies of IPTG-induced *E. coli* BL21(DE3) lysate with pET-30a-ΔN33Cap; lane 6 and 7: purified ΔN33Cap protein. Lanes 1 to 6 are for the Western blot of anti-His-Tag, and Lane 7 is for the Western blot of positive serum.

**Figure 2 viruses-17-01340-f002:**
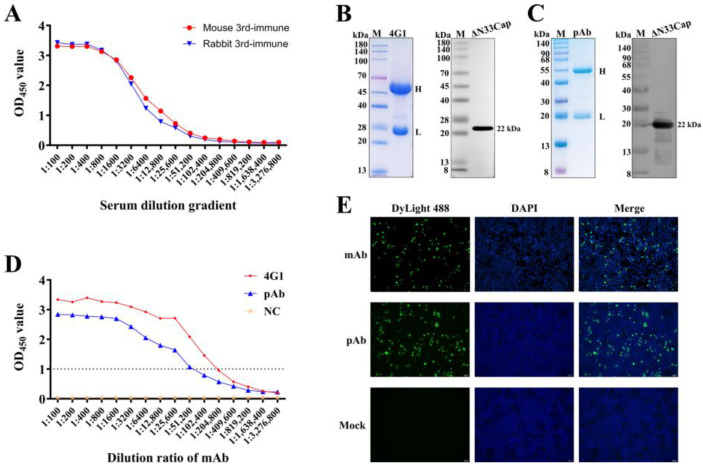
Preparation and characterization of mAb and pAb. (**A**): Serum antibody titers of immunized mice and rabbits. (**B**): SDS-PAGE and Western blot validation of mAb. Western blot of the ΔN33Cap protein using the mAb as the primary antibody and HRP-conjugated goat anti-mouse IgG as secondary antibody. (**C**): SDS-PAGE and Western blot validation of pAb. Western blot of the ΔN33Cap protein using the pAb as the primary antibody and HRP-conjugated goat anti-rabbit IgG as secondary antibody. (**D**): mAb and pAb titers were determined by indirect ELISA. (**E**): IFA of mAb and pAb affinity for HA-Cap in HEK-293T cells. Green fluorescence indicates specific mAb and pAb binding to HA-Cap; Blue fluorescence labels cell nuclei. DyLight488 is goat anti-mouse IgG(H&L) and goat anti-rabbit IgG(H&L).

**Figure 3 viruses-17-01340-f003:**
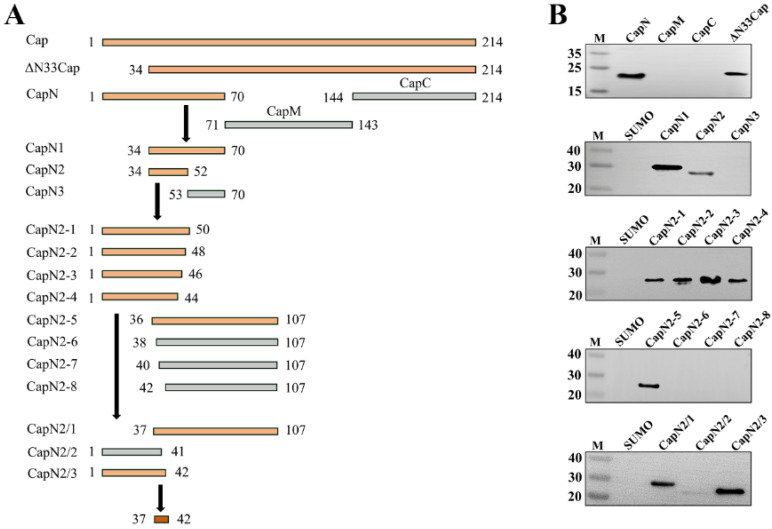
Fine mapping of the linear epitope recognized by mAb. (**A**): Procedure for detecting truncated PCV3 Cap protein. orange legends represent mAb-reactive fragments; gray legends represent mAb-nonreactive fragments; red legend represents minimal linear epitope. (**B**): Western blot of truncated PCV3 Cap protein according to the detection procedure.

**Figure 4 viruses-17-01340-f004:**
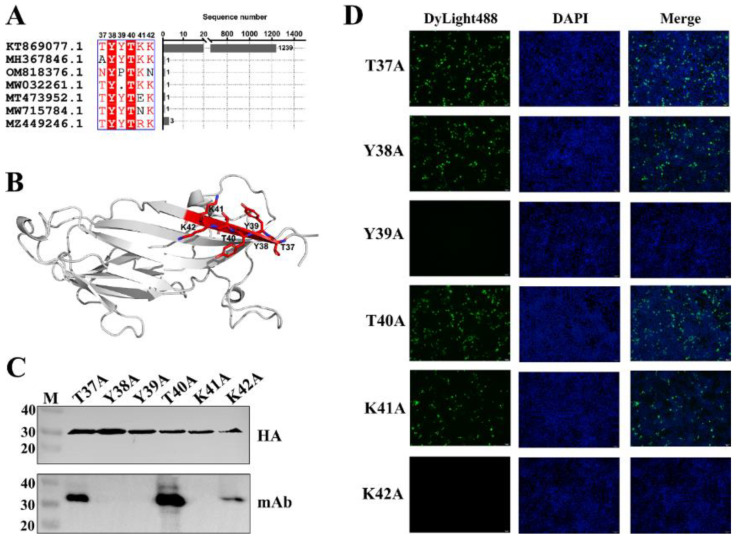
Conservation, location, and key amino acids identified in the epitopes. (**A**): Conservation analysis of the minimal linear epitope recognized by mAb. (**B**): Localization of the minimal linear epitope in the structure of the Cap protein is marked in red. (**C**): Western blot of sequential point mutations in the epitope. (**D**): IFA of sequential point mutations in the epitope. Amino acids in the core epitope were mutated to alanine in turn and annotated above the lane.

**Figure 5 viruses-17-01340-f005:**
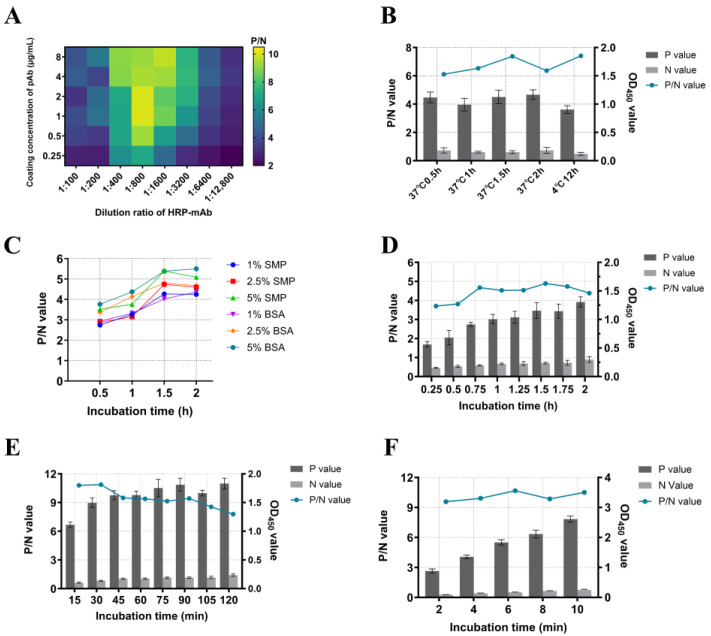
Optimization of reaction conditions for DAS-ELISA. (**A**): The optimal matching of the pAb coating concentration and the HRP-mAb working dilution ratio, P/N values are displayed in the heat map as color shades. (**B**): The optimal pAb coating conditions. (**C**): The optimal blocking solution and optimal blocking time; (**D**): The optimal incubation time of serum; (**E**): The optimal incubation time of HRP-mAb; (**F**): The optimal color development time of TMB substrates. Positive values and negative values are presented in a bar chart with bar values; P/N values are presented in a line chart.

**Figure 6 viruses-17-01340-f006:**
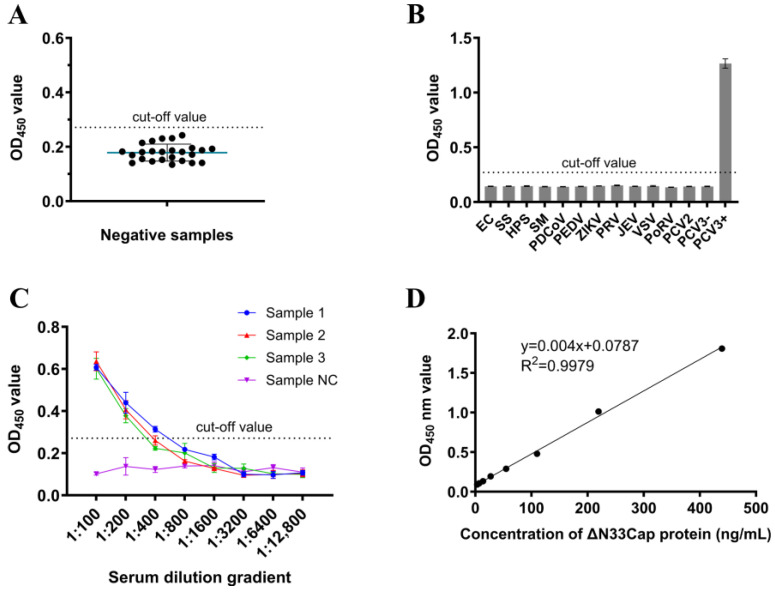
Performance validation of DAS-ELISA. (**A**) Negative sera were tested using DAS-ELISA for calculating the cut-off value. The cut-off value is $\bar{X}$ + 3 SD. (**B**): Specificity test of DAS-ELISA. (**C**): Sensitivity test of DAS-ELISA. (**D**): The linear standard curve for detecting Cap protein using DAS-ELISA.

**Table 1 viruses-17-01340-t001:** Intra-batch repetitive test.

Sample	OD_450_			Mean Value	Standard Deviation	Coefficient of Variation
1	1.354	1.435	1.377	1.389	0.0417	3.01%
2	0.777	0.858	0.885	0.840	0.0562	6.69%
3	0.269	0.265	0.254	0.263	0.0078	2.96%
4	0.517	0.572	0.51	0.533	0.034	6.37%

**Table 2 viruses-17-01340-t002:** Inter-batch repetitive test.

Sample	OD_450_			Mean Value	Standard Deviation	Coefficient of Variation
1	1.393	1.331	1.317	1.347	0.0404	3.00%
2	0.823	0.812	0.799	0.8113	0.012	1.48%
3	0.21	0.206	0.215	0.2103	0.0045	2.14%
4	0.591	0.58	0.647	0.606	0.0359	5.93%

**Table 3 viruses-17-01340-t003:** Comparison of DAS-ELISA and qPCR for the detection of PCV3 in serum.

		DAS-ELISA				
		Positive	Negative	Total	Coincidence rate	Kappa value
qPCR	Positive	51	3	54	93.33%	0.837
	Negative	2	19	21		
	Total	55	10	75		

## Data Availability

All the data generated during the current study are included in the manuscript.

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
