# Peer review of "Development of a Highly Specific Monoclonal Antibody-Based Sandwich ELISA for Rapid Detection of Porcine Circovirus Type 3"

_viruses, 2025, doi:10.3390/v17101340_

Round 1
Reviewer 1 Report
Comments and Suggestions for Authors
The pandemic of the porcine circovirus type 3 (PCV3) has caused huge economic losses. So, much attention has been paid to PCV3 detection. In this study, a sandwich ELISA was developed for the efficient and sensitive detection of PCV3 by using rabbit polyclonal antibody (pAb) as the capture antibody and mAb 4G1 as the detection antibody. The ELISA exhibits good specificity and sensitivity, with a limit of detection for the Cap protein of 3.4 ng/mL. The work is interesting and meaningful. However, before it is accepted for publication, the review still has some comments and suggestions as follows:
Statistical analysis of the amino acid sequence alignment of PCV3 Cap reveals that the hexapeptide segment is highly conserved. I cannot agree with authors according Fig4A, more protein sequences should be classified and listed according to genotypes PCV3a-1, PCV3a-2 and PCV3b. Additional experiments to confirm cross-genotype reactivity should be performed.
The ELISA exhibits good sensitivity to Cap protein, is there a well correlation between virus CT value and OD value like these in major protein of PCV3? And a total of 75 serum samples were tested by qPCR and DAS-ELISA, The overall agreement between the two methods was 93.33% (70/75), are there any other methods to further confirm the inconsistent results of the 5 samples (the background of the serums)?
Fig 1 Western blot analysis using an anti-6×His tag monoclonal antibody confirmed the protein size, positive serum sample should be performed to confirm the immunogenicity of expressed proteins.
Fig 2c mAb titer was determined by indirect ELISA, maybe it’s better to test the HRP-mAb titer by direct ELISA. In addition, the cut off lines must be added in Fig 2A, C, Fig5A, C and Fig7B, C.
The authors should be more careful with citations and use papers that were the first to report. e.g., “Self-assembly into virus-like particles of the recombinant capsid protein of porcine circovirus type 3 and its application on antibodies detection” and “A Sandwich ELISA for Quality Control of PCV2 Virus-like Particles Vaccine”
Lines 301 More information should be provided for negative samples and various strains of the virus, e.g., sources and ages.
The experimental steps for HRP labed mAb should be described in the materials and methods, and the positive and negative samples used in systematically optimized to analysis the maximum positive-to-negative (P/N) ratio should be described.
Author Response
Response to Reviewer 1 Comments
Dear Editor and Reviewer,
We sincerely appreciate the time and effort you have dedicated to reviewing our manuscript. We are grateful for the constructive comments and suggestions provided by the reviewers, which have significantly improved the quality of our work. Below, we provide a point-by-point response to the reviewers' comments. We will use the revision mode to modify the comments in the word manuscript and mark the modified positions in detail. All changes in the manuscript are highlighted in red for easy reference.
Comments 1: Statistical analysis of the amino acid sequence alignment of PCV3 Cap reveals that the hexapeptide segment is highly conserved. I cannot agree with authors according Fig4A, more protein sequences should be classified and listed according to genotypes PCV3a-1, PCV3a-2 and PCV3b. Additional experiments to confirm cross-genotype reactivity should be performed.
Response 1: We appreciate the reviewer's insightful comments. As the reviewer pointed out, it would indeed be better to conduct the conservation analysis of the hexapeptide amino acid sequence among different PCV3 genotypes. We also considered this issue. However, given the current lack of a uniformly accepted genotyping scheme for PCV3, we opted to cautiously analyze 1,247 sequences. It is worth noting that these sequences were downloaded from GenBank and include those uploaded from 2016 to December 2024, covering the genotypes mentioned by the reviewer, such as PCV3a-1, PCV3a-2, and PCV3a-b. To ensure better understanding for the readers and reviewers, we have decided to include a more detailed description of the conservation analysis in the Materials and Methods subsection 2.5 (page 4, lines 172–177) of the manuscript and provide the sequence information in the supplementary materials (Supplement S1-sheet12), which is described as: “In parallel, all available PCV3 genome sequences uploaded to GenBank as of 2024 (Supplement S1) were retrieved. Multiple sequence alignment was performed using MEGA11 to assess the conservation of the core epitope. The predicted Cap protein structure was then visualized with PyMOL, and the location of the core epitope was mapped and labeled within the structural model.”
Comments 2: The ELISA exhibits good sensitivity to Cap protein, is there a well correlation between virus CT value and OD value like these in major protein of PCV3? And a total of 75 serum samples were tested by qPCR and DAS-ELISA, The overall agreement between the two methods was 93.33% (70/75), are there any other methods to further confirm the inconsistent results of the 5 samples (the background of the serums)?
Response 2: We thank the reviewer for raising this interesting and important point regarding the correlation between viral load (as indicated by qPCR CT values) and the level of Cap protein antigenemia (as indicated by DAS-ELISA OD values). In this study, we did observe a general trend where samples with lower CT values (indicating higher viral DNA load) tended to yield higher OD450 values in our DAS-ELISA. This suggests a positive correlation between the presence of the viral genome and the detection of the Cap protein antigen. However, it is important to note that clinical PCV3 infection is more complex. Factors such as the degradation of both PCV3 antigen and nucleic acid, as well as the recurrent nature of PCV3 infections, may influence detection results. Moreover, the peak of viremia (DNA detection) and antigenemia (protein detection) may not occur simultaneously during the course of infection. Therefore, the observations from the clinical serum samples we tested may not be representative or generalizable. A formal and rigorous correlation analysis between CT and OD values would require experimental PCV3 infection studies with longitudinal monitoring, or large-scale testing of clinical samples. However, a formal and rigorous correlation analysis was not the primary focus of this methodological paper, which aimed first to establish and validate the novel DAS-ELISA itself.
Comments 3: Fig 1 Western blot analysis using an anti-6×His tag monoclonal antibody confirmed the protein size, positive serum sample should be performed to confirm the immunogenicity of expressed proteins.
Response 3: We agree with the reviewer's comment and have incorporated this result figure into subsection 3.1 of the Results section (page 6, Lines 229; Fig. 1B). And modified the figure note as follows: "lane 6 and 7: purified ΔN33Cap protein. Lanes 1 to 6 are for the Western Blot of anti-His-Tag, and Lane 7 is for the Western Blot of positive serum (page 6, Lines 236-237)."
Comments 4: Fig 2c mAb titer was determined by indirect ELISA, maybe it’s better to test the HRP-mAb titer by direct ELISA. In addition, the cut off lines must be added in Fig 2A, C, Fig5A, C and Fig7B, C.
Response 4: We thank the reviewer for this valuable suggestion. The reviewer is correct that a direct ELISA using HRP-conjugated mAb can be an excellent method for precise titer determination. In the current study, we employed an indirect ELISA for the initial screening and titer measurement of hybridoma supernatants for the practical and methodological reasons. During the early stages of hybridoma screening, we are dealing with a large number of candidate clones. An indirect ELISA allows us to rapidly screen all supernatants using a single, universal HRP-conjugated secondary antibody. Our indirect ELISA protocol was rigorously optimized to minimize background and non-specific binding from the secondary antibody. The high signal-to-noise ratio observed in our results (Fig. 2C) confirms the specificity and reliability of the assay for comparing relative antibody titers across samples. Since all samples were analyzed under identical conditions using the same secondary reagent, the relative titers obtained are consistent and valid for comparing the binding strength of the different mAbs produced by various clones. Furthermore, as discussed in the manuscript (Page 12, Lines 367-368), while direct ELISA could be used to distinguish sensitivity differences among mAbs targeting different epitopes in detection applications, our antibody additivity assay performed using indirect ELISA (Supplementary Table S4) indicated that all four selected mAbs recognize the same epitope. Therefore, to maintain methodological consistency throughout the entire antibody development process, we chose to use indirect ELISA as the standard approach.
We agree with the reviewers' opinions regarding the result figures. We have added clear cut-off lines to Figures 7B and 7C (page 11, lines 335-336). This addition provides a definitive threshold for distinguishing positive from negative results in our DAS-ELISA validation assays, significantly enhancing the clarity of these figures. Regarding Figures 2A, 2C, 5A, and 5C (which depict antibody titer curves from immunized mice and rabbits, the titer curves of purified mAb and pAb), we would like to provide a clarification based on standard immunological presentation. These panels are intended to quantitatively illustrate the dynamic range of the immune response and are used to calculate the antibody titer value. The titer itself is a quantitative measure defined as the reciprocal of the highest dilution that yields a signal significantly above background. In such quantitative graphical representations of titration data, it is not the conventional practice to include a horizontal cut-off line, as the entire curve illustrates the transition from positive to negative, and the derived titer value inherently contains the cut-off logic. Adding a line could be redundant and potentially confusing for the reader. Meanwhile, we added negative controls during the measurement to ensure the rigor of the experiment.
Comments 5: The authors should be more careful with citations and use papers that were the first to report. e.g., “Self-assembly into virus-like particles of the recombinant capsid protein of porcine circovirus type 3 and its application on antibodies detection” and “A Sandwich ELISA for Quality Control of PCV2 Virus-like Particles Vaccine”
Response 5: We agree with the reviewer's comment and sincerely apologize for our oversight. We believe it is entirely reasonable to use the papers that were the first to report the findings as our references. Accordingly, we have made the following changes: Delete the original reference papers. Citation of paper “Self-assembly into virus–like particles of the recombinant capsid protein of porcine circovirus type 3 and its application on antibodies detection” has been included as reference [31] (listed on page 15, lines 531–533). Citation of paper “A Sandwich ELISA for Quality Control of PCV2 Virus-like Particles Vaccine” has been included as reference [34] (listed on page 16, lines 539–540).
Comments 6: Lines 301 More information should be provided for negative samples and various strains of the virus, e.g., sources and ages.
Response 6: We agree with the reviewer's comment. Regarding the information on negative samples, we have added the following sentence in subsection 2.6 (page 5, lines 194-195) of the Materials and Methods: "The positive and negative serum were gifted from Wuhan Keqian Biology Co., Ltd." Concerning the details of the bacterial and viral strains, we have revised the description originally on lines 199-200 of page 5 to now read: "Specificity was evaluated against cross-reactive pathogens that were collected and stored in our lab, including: Escherichia coli (EC), …."
Comments 7: The experimental steps for HRP labed mAb should be described in the materials and methods, and the positive and negative samples used in systematically optimized to analysis the maximum positive-to-negative (P/N) ratio should be described.
Response 7: We agree with the reviewer’s comment and apologize for our oversight. We have added the following content to the manuscript: In subsection 2.6 of the Materials and Methods (page 4, lines 183–185), the experimental procedure for the HRP-labeled mAb has been added: “The mAb 4G1 was conjugated with HRP using the periodate method, according to the manufacturer’s instructions (Sangon, Japan).” Additionally, in subection 2.6 (page 5, lines194–195), a description of the positive and negative samples has been included: “The positive and negative serum used for system optimization were provided by Wuhan Keqian Biology Co., Ltd.”
Finally, based on the reviewer's constructive feedback, we believe it is necessary to add a paragraph to the discussion to address the concerns raised, enhance readers' understanding of the subject and content of this study, and provide some forward-looking perspectives. The specific content (page 13, lines 410-442) is as follows: “Given that PCV3 is detectable in multiple tissues of infected pigs and sporadically in oral fluids, feces, nasal swabs, semen, and colostrum—coupled with evidence of persistent viremia post-infection (8, 29). Therefore, serum was selected as the detection target in this study to maximize detection efficiency. To evaluate the performance of the DAS-ELISA, we tested randomly collected serum samples from clinically healthy pigs. In the absence of commercial kits for direct comparison, samples were initially screened using qPCR to detect PCV3 nucleic acid, followed by antigen detection using the DAS-ELISA, to assess the concordance between the two methods. The comparison with qPCR, a commonly used diagnostic tool in swine farms, serves as a preliminary indication of the detection capability of the DAS-ELISA. This comparison was not to establish a direct correlation but to use qPCR, the most widely adopted and sensitive method for detecting PCV3 genetic material, as a benchmark for an initial clinical performance evaluation of our assay (14, 45). A general trend was observed wherein samples with lower Ct values (indicating higher viral DNA load) tended to yield higher ODâ‚„â‚…â‚€ values in the DAS-ELISA, suggesting a positive association between viral genome detection and Cap protein antigen levels. However, PCV3 can be transmitted both horizontally and vertically, while its infection, replication, nucleic acid degradation, and antigen degradation under field conditions exhibit a cyclical and recurrent pattern (46). Moreover, the peaks of viremia (DNA detection) and antigenemia (protein detection) may not occur simultaneously during the course of infection (47). Consequently, the relationship between nucleic acid and antigen levels is difficult to fully elucidate using cross-sectional clinical serum samples alone. The observations derived from this sample set may not be fully representative or generalizable, and the primary aim of this analysis was to validate the DAS-ELISA itself. We did observe a formal and rigorous correlation analysis between Ct and OD values would require longitudinal monitoring in experimental PCV3 infection studies. For field-level investigations, large-scale testing of clinical samples is needed, and the relationship between these values may be influenced by factors such as the age and health status of the pigs. However, such an analysis first requires establishing the kinetics of PCV3 nucleic acid and antigen presence through experimental infection to define the detection window. This will be the main objective of our follow-up work. In summary, these findings provide a powerful tool for exploring the dynamics of antigen during PCV3 replication and offer a valuable reference for future diagnostic development.”
Reviewer 2 Report
Comments and Suggestions for Authors
This manuscript describes the development of a double antibody sandwich ELISA (DAS-ELISA) for detecting porcine circovirus 3 (PCV3). In this work, a monoclonal antibody (mAb 4G1) against PCV3 Cap protein was prepared and used as detecting antibody, and rabbit polyclonal antibody (pAb) was applied as the capture antibody. Meanwhile, the reaction conditions for DAS-ELISA were determined and optimized, which meet with the need for monitoring PCV3 infection in swine herds. Overall, this study is complete with novelty.
Major comment:
The authors compared the DAS-ELISA with qPCR for the detection of PCV3 in sera. This is a unreasonable test. DAS-ELISA is a method for antigen (protein) detection, whereas qPCR is a technique for viral nucleoid acid detection. A method for detecting viral protein (such as WB) or virus isolation should be applied.
Minor comment:
The results section is redundant, and needs to be concise and reorganized. Some figures can be removed, such as figure. 2A and 2C,5A and 5C, and whole figure 6. Table 1 and table 2 can be removed.
Author Response
Response to Reviewer 2 Comments
Dear Editor and Reviewer,
We sincerely appreciate the time and effort you have dedicated to reviewing our manuscript. We are grateful for the constructive comments and suggestions provided by the reviewers, which have significantly improved the quality of our work. Below, we provide a point-by-point response to the reviewers' comments. We will use the revision mode to modify the comments in the word manuscript and mark the modified positions in detail. All changes in the manuscript are highlighted in red for easy reference.
Comments 1: The authors compared the DAS-ELISA with qPCR for the detection of PCV3 in sera. This is an unreasonable test. DAS-ELISA is a method for antigen (protein) detection, whereas qPCR is a technique for viral nucleoid acid detection. A method for detecting viral protein (such as WB) or virus isolation should be applied.
Response 1: We appreciate the reviewer's insightful comments. We agree with the reviewer that qPCR and DAS-ELISA target different molecular entities (nucleic acid vs. protein) and are therefore not directly comparable. Our intention in conducting this comparison was not to establish a direct correlation but to use qPCR — as the most widely adopted and sensitive method for detecting the presence of PCV3 genetic material — as a benchmark for an initial clinical performance assessment of our assay. This approach is commonly employed in the early validation of new diagnostic tests to evaluate their performance against a current gold standard using clinical samples. The high concordance rate (93.33%) we observed indicates that our DAS-ELISA successfully detects the virus in samples where it is molecularly identified.
We also agree that virus isolation would be a valuable comparator. However, the efficient isolation of PCV3 in vitro remains a significant challenge for the field. There is currently no robust, standardized, or widely available cell culture system for this virus, which precluded the use of virus isolation for validating clinical samples in this study. Regarding the suggestion of Western Blot (WB), While WB is undoubtedly a powerful technique for protein detection, we concluded that a direct comparison between WB and DAS-ELISA on clinical sera may not be definitive due to fundamental methodological differences. The two assays have vastly different antigen capture dynamics: The capture is performed by pAbs immobilized on a plate in our DAS-ELISA, whereas in WB, proteins are denatured, linearly separated by electrophoresis, and non-specifically transferred into a PVDF membrane.
Furthermore, we believe the most rigorous validation of our antigen-specific DAS-ELISA is demonstrated through its internal validation metrics, which we have comprehensively provided:
- Exceptional Specificity: No cross-reactivity with a panel of other major swine pathogens.
- High Sensitivity: A low detection limit for the Cap protein (3.4 ng/mL) and positive detection in sera diluted up to 1:800.
- High Precision: Low intra- and inter-assay coefficients of variation (<10%).
- Use of Characterized Reagents: The assay is built upon a well-characterized monoclonal antibody targeting a highly conserved linear epitope.
We hope this explanation clarifies our experimental design and validation approach.
Comments 2: The results section is redundant, and needs to be concise and reorganized. Some figures can be removed, such as figure. 2A and 2C,5A and 5C, and whole figure 6. Table 1 and table 2 can be removed.
Response 2: We appreciate the reviewer's comments. In our study, the primary focus lies in antibody preparation and method development. We consider these data and results to be an integral part of our research, as the provided figures and tables further demonstrate the scientific validity and reliability of both the antibodies produced and the DAS-ELISA method established in this work. This manner of presentation, integrating validation data within the main study, has also been employed in previous publications such as: “Wang J, Lei B, Zhang W, Li L, Ji J, Liu M, et al. Preparation of monoclonal antibodies against the capsid protein and development of an epitope-blocking enzyme-linked immunosorbent assay for detection of the antibody against porcine circovirus 3. Animals (Basel). 2024;14.”
Finally, based on the reviewer's constructive feedback, we believe it is necessary to add a paragraph to the discussion to address the concerns raised, enhance readers' understanding of the subject and content of this study, and provide some forward-looking perspectives. The specific content (page 13, lines 410-442) is as follows: “Given that PCV3 is detectable in multiple tissues of infected pigs and sporadically in oral fluids, feces, nasal swabs, semen, and colostrum—coupled with evidence of persistent viremia post-infection (8, 29). Therefore, serum was selected as the detection target in this study to maximize detection efficiency. To evaluate the performance of the DAS-ELISA, we tested randomly collected serum samples from clinically healthy pigs. In the absence of commercial kits for direct comparison, samples were initially screened using qPCR to detect PCV3 nucleic acid, followed by antigen detection using the DAS-ELISA, to assess the concordance between the two methods. The comparison with qPCR, a commonly used diagnostic tool in swine farms, serves as a preliminary indication of the detection capability of the DAS-ELISA. This comparison was not to establish a direct correlation but to use qPCR, the most widely adopted and sensitive method for detecting PCV3 genetic material, as a benchmark for an initial clinical performance evaluation of our assay (14, 45). A general trend was observed wherein samples with lower Ct values (indicating higher viral DNA load) tended to yield higher ODâ‚„â‚…â‚€ values in the DAS-ELISA, suggesting a positive association between viral genome detection and Cap protein antigen levels. However, PCV3 can be transmitted both horizontally and vertically, while its infection, replication, nucleic acid degradation, and antigen degradation under field conditions exhibit a cyclical and recurrent pattern (46). Moreover, the peaks of viremia (DNA detection) and antigenemia (protein detection) may not occur simultaneously during the course of infection (47). Consequently, the relationship between nucleic acid and antigen levels is difficult to fully elucidate using cross-sectional clinical serum samples alone. The observations derived from this sample set may not be fully representative or generalizable, and the primary aim of this analysis was to validate the DAS-ELISA itself. We did observe a formal and rigorous correlation analysis between Ct and OD values would require longitudinal monitoring in experimental PCV3 infection studies. For field-level investigations, large-scale testing of clinical samples is needed, and the relationship between these values may be influenced by factors such as the age and health status of the pigs. However, such an analysis first requires establishing the kinetics of PCV3 nucleic acid and antigen presence through experimental infection to define the detection window. This will be the main objective of our follow-up work. In summary, these findings provide a powerful tool for exploring the dynamics of antigen during PCV3 replication and offer a valuable reference for future diagnostic development.”
Reviewer 3 Report
Comments and Suggestions for Authors
Review of the Manuscript:
"Development of a Highly Specific Monoclonal Antibody-Based Sandwich ELISA for Rapid Detection of Porcine Circovirus Type 3"
In this study, the authors developed a monoclonal antibody (mAb 4G1) targeting a linear epitope consisting of six amino acids within the Cap protein of PCV3. Based on the high conservation of this epitope, they established a double-antibody sandwich ELISA (DAS-ELISA) to monitor the presence of PCV3 in swine herds.
Overall, I consider the ELISA presented in this study to be a valuable diagnostic tool for detecting PCV3 in field conditions.
Suggestions to Improve the Quality of the Study:
- Introduction Section
- I recommend including more information on the pathogenesis of PCV3 in pigs. Specifically:
- Highlight the main sources of the virus.
- Describe the types of samples most commonly used for diagnosis.
- Discuss the window of antigen detectability during infection.
This context is essential for understanding the potential sample types suitable for this ELISA.
- Methods Section
- Please provide more details about the 75 serum samples used to compare DAS-ELISA with qPCR:
- What was the clinical status of the pigs?
- Where and under what conditions were the samples collected?
- Results Section
- The comparison between DAS-ELISA and qPCR could be strengthened with an additional experiment:
- Use serial ten-fold dilutions of a viral antigen with a known titer.
- Perform both tests on these dilutions until both yield negative results.
This would contextualize the sensitivity and detection limits of each method.
- Regarding the evaluation of clinical serum samples:
- What were the Ct values for samples that tested positive by both methods?
- For the three discordant samples (qPCR-positive, ELISA-negative), what were their Ct values?
- Could the high agreement between methods be influenced by the clinical status of the animals?
- How did the authors ensure that the clinical samples represented the full spectrum of antigen levels typically found during natural infection?
Ideally, this comparison should include samples collected at various time points post-infection from experimentally infected pigs.
- Discussion Section
- Please expand the discussion to incorporate the results of the proposed experiments and address the limitations and implications of the current findings.
Author Response
Response to Reviewer 3 Comments
Dear Editor and Reviewer,
We sincerely appreciate the time and effort you have dedicated to reviewing our manuscript. We are grateful for the constructive comments and suggestions provided by the reviewers, which have significantly improved the quality of our work. Below, we provide a point-by-point response to the reviewers' comments. We will use the revision mode to modify the comments in the word manuscript and mark the modified positions in detail. All changes in the manuscript are highlighted in red for easy reference.
Comments 1: Introduction Section. I recommend including more information on the pathogenesis of PCV3 in pigs. Specifically: Highlight the main sources of the virus. Describe the types of samples most commonly used for diagnosis. Discuss the window of antigen detectability during infection. This context is essential for understanding the potential sample types suitable for this ELISA.
Response 1: We agree with the reviewer's comment. We have added the following statement to the Introduction section of our manuscript (page 2, lines 50–54): " In addition, PCV3 can be transmitted both horizontally and vertically. The infection leads to persistent viremia, with intermittent viral shedding observed in feces, nasal secretions, semen, and colostrum (10, 11). Moreover, PCV3 nucleic acid can be widely detected in various tissues, including the heart, liver, spleen, lungs, kidneys, intestines, and lymph nodes (12)." This addition serves to highlight potential sources of the virus and sample types suitable for diagnostic purposes. To the best of our knowledge, there are currently no reports on antigen detection during PCV3 infection. Therefore, we have moved the interpretation related to this aspect to the Discussion section.
Comments 2: Methods Section. Please provide more details about the 75 serum samples used to compare DAS-ELISA with qPCR: What was the clinical status of the pigs? Where and under what conditions were the samples collected?
Response 2: We thank the reviewer for this valuable suggestion. On lines 211–215 of page 5, we have added the following statement: " To evaluate the performance of the DAS-ELISA, a comparative analysis was performed on 75 routine serum samples collected from healthy pigs on a farm in Guangzhou. All samples were first tested for PCV3 nucleic acid by qPCR and subsequently for PCV3 antigen by DAS-ELISA. The agreement between the two methods, quantified by the kappa value, was calculated using Microsoft Excel."
Comments 3: Results Section. The comparison between DAS-ELISA and qPCR could be strengthened with an additional experiment: Use serial ten-fold dilutions of a viral antigen with a known titer. Perform both tests on these dilutions until both yield negative results. This would contextualize the sensitivity and detection limits of each method. Regarding the evaluation of clinical serum samples: What were the Ct values for samples that tested positive by both methods? For the three discordant samples (qPCR-positive, ELISA-negative), what were their Ct values? Could the high agreement between methods be influenced by the clinical status of the animals? How did the authors ensure that the clinical samples represented the full spectrum of antigen levels typically found during natural infection? Ideally, this comparison should include samples collected at various time points post-infection from experimentally infected pigs.
Response 3: We thank the reviewer for these insightful comments. We have carefully considered the points raised and provide the following responses based on our current study design and data:
In this study, the data from the clinical samples tested by both qPCR and DAS-ELISA are provided in Supplementary File S1. We did observe a general trend where samples with lower CT values (indicating higher viral DNA load) tended to yield higher OD450 values. This suggests a positive correlation between the presence of the viral genome and the detection of the Cap protein antigen. However, it is important to note that clinical PCV3 infection is more complex. PCV3 can be transmitted both horizontally and vertically, while its infection, replication, nucleic acid degradation, and antigen degradation under field conditions exhibit a cyclical and recurrent pattern. Moreover, the peaks of viremia (DNA detection) and antigenemia (protein detection) may not occur simultaneously during the course of infection. Therefore, using clinical serum samples alone makes it difficult to fully understand the relationship between nucleic acid and antigen levels post-PCV3 infection. Consequently, the observations from the clinical serum samples we tested may not be representative or generalizable. Additionally, our intention in conducting this comparison was not to establish a direct correlation but to use qPCR — as the most widely adopted and sensitive method for detecting the presence of PCV3 genetic material — as a benchmark for an initial clinical performance assessment of our assay.
As rightly noted by the reviewer, a formal and rigorous correlation analysis between Ct and OD values requires longitudinal monitoring in experimental PCV3 infection studies. For investigation at the clinical level, large-scale testing of clinical samples is necessary, and the relationship between these values may be influenced by the age and health status of the pigs. However, this would first require establishing the dynamics of PCV3 nucleic acid and antigen presence through experimental infection studies to define the detection window. Nevertheless, such a formal and strict correlation analysis was not the primary focus of this methodological paper, which aimed first and foremost to establish and validate the novel DAS-ELISA assay itself. However, we have still taken the reviewer's suggestion seriously and have incorporated a discussion of this limitation and its implications for clinical application into the Discussion section of the revised manuscript. We will also address this point directly in our response to Comment 4.
Comments 4: Discussion Section. Please expand the discussion to incorporate the results of the proposed experiments and address the limitations and implications of the current findings.
Response 4: We thank the reviewer for insightful comment and agree on the necessity to address this point. In conjunction with our response to Comment 3, we have provided the following clarification in the Discussion section and added it to page 13, lines 410-442: “Given that PCV3 is detectable in multiple tissues of infected pigs and sporadically in oral fluids, feces, nasal swabs, semen, and colostrum—coupled with evidence of persistent viremia post-infection (8, 29). Therefore, serum was selected as the detection target in this study to maximize detection efficiency. To evaluate the performance of the DAS-ELISA, we tested randomly collected serum samples from clinically healthy pigs. In the absence of commercial kits for direct comparison, samples were initially screened using qPCR to detect PCV3 nucleic acid, followed by antigen detection using the DAS-ELISA, to assess the concordance between the two methods. The comparison with qPCR, a commonly used diagnostic tool in swine farms, serves as a preliminary indication of the detection capability of the DAS-ELISA. This comparison was not to establish a direct correlation but to use qPCR, the most widely adopted and sensitive method for detecting PCV3 genetic material, as a benchmark for an initial clinical performance evaluation of our assay (14, 45). A general trend was observed wherein samples with lower Ct values (indicating higher viral DNA load) tended to yield higher ODâ‚„â‚…â‚€ values in the DAS-ELISA, suggesting a positive association between viral genome detection and Cap protein antigen levels. However, PCV3 can be transmitted both horizontally and vertically, while its infection, replication, nucleic acid degradation, and antigen degradation under field conditions exhibit a cyclical and recurrent pattern (46). Moreover, the peaks of viremia (DNA detection) and antigenemia (protein detection) may not occur simultaneously during the course of infection (47). Consequently, the relationship between nucleic acid and antigen levels is difficult to fully elucidate using cross-sectional clinical serum samples alone. The observations derived from this sample set may not be fully representative or generalizable, and the primary aim of this analysis was to validate the DAS-ELISA itself. We did observe a formal and rigorous correlation analysis between Ct and OD values would require longitudinal monitoring in experimental PCV3 infection studies. For field-level investigations, large-scale testing of clinical samples is needed, and the relationship between these values may be influenced by factors such as the age and health status of the pigs. However, such an analysis first requires establishing the kinetics of PCV3 nucleic acid and antigen presence through experimental infection to define the detection window. This will be the main objective of our follow-up work. In summary, these findings provide a powerful tool for exploring the dynamics of antigen during PCV3 replication and offer a valuable reference for future diagnostic development.”
We hope that the added discussion section will address the reviewer's concerns, provide readers with a better understanding of the themes and content of this study, and offer some forward-looking perspectives.
Round 2
Reviewer 2 Report
Comments and Suggestions for Authors
The authors basically address the comments based on their explanations, not additional tests.
Reviewer 3 Report
Comments and Suggestions for Authors
I like to thank the authors for their responses, at this point I don't have more concerns about this manuscript.